# Tumor Suppressor p53 Inhibits Hepatitis B Virus Replication by Downregulating HBx via E6AP-Mediated Proteasomal Degradation in Human Hepatocellular Carcinoma Cell Lines

**DOI:** 10.3390/v14102313

**Published:** 2022-10-21

**Authors:** Ha-Yeon Lim, Jiwoo Han, Hyunyoung Yoon, Kyung Lib Jang

**Affiliations:** 1Department of Integrated Biological Science, The Graduate School, Pusan National University, Busan 46241, Korea; 2Department of Microbiology, College of Natural Science, Pusan National University, Busan 46241, Korea; 3Microbiological Resource Research Institute, Pusan National University, Busan 46241, Korea

**Keywords:** E6-associated protein, hepatitis B virus, proteasome, HBx, p53

## Abstract

HBx, a multifunctional regulatory protein, plays an essential role in the replication and pathogenesis of the hepatitis B virus (HBV). In this study, we found that in human hepatoma cells, the tumor suppressor p53 downregulates HBx via ubiquitin-dependent proteasomal degradation. p53 transcriptional activity that results from HBV infection was not essential for this effect. This was shown by treatment with a potent p53 inhibitor, pifithrin-α. Instead, we found that p53 facilitated the binding of E6-associated protein (E6AP), which is an E3 ligase, to HBx and induced E6AP-mediated HBx ubiquitination in a ternary complex of p53, E6AP, and HBx. The ability of p53 to induce E6AP-mediated downregulation of HBx and inhibit HBV replication was demonstrated in an in vitro HBV infection system. This study may provide insights into the regulation of HBx and HBV replication, especially with respect to p53 status, which may also help in understanding HBV-associated tumorigenesis in patients.

## 1. Introduction

Hepatitis B virus (HBV) is a major human hepatotropic pathogen that causes acute and chronic hepatitis, liver cirrhosis, and hepatocellular carcinoma (HCC) [1,2]. As a member of the *Hepadnaviridae* family, HBV encapsidates a partially double-stranded circular DNA genome of approximately 3.2 kbp via reverse transcription of pregenomic RNA [2,3]. Among the four open reading frames, S, C, P, and X, in the HBV genome, the smallest, X, encodes a 17 kDa regulatory protein named HBV X protein (HBx). This multifunctional protein has attracted particular attention because it is implicated as a viral oncoprotein in HBV-mediated HCC development [1,4]. In addition, HBx has been demonstrated to be a positive regulator of HBV replication in several experimental systems, including human hepatocyte chimeric mice [5], HBV-transgenic mice [6], murine hydrodynamic injection [7], and in vitro infection models [7,8]. HBx stimulates four viral promoters to synthesize HBV mRNA and pre-genomic RNA from a covalently closed circular DNA template [2,9,10]. HBx also contributes to HBV replication indirectly by deregulating cellular signaling pathways, such as the cytosolic calcium signaling pathway [11] and the phosphatidylinositol 3-kinase/Akt pathway [12]. While our knowledge on the role of HBx as a positive regulator of HBV replication is expanding, the underlying mechanism of HBx regulation during HBV replication remains limited.

The ubiquitin (Ub)–proteasome system (UPS) serves as an anti-viral defense system by facilitating degradation of viral proteins. It also acts against HBV infection, and degrades several viral proteins, including HBx [13]. Early studies have shown that HBx is present at an extremely low level in patients with chronic hepatitis and HBV infection because it is rapidly degraded by the UPS in the host cells [14,15]. Further studies have identified several cellular proteins that limit HBV replication by downregulating HBx via UPS. Seven in absentia homologue 1 (Siah-1), an E3 ligase, can induce Ub-dependent proteasomal degradation of HBx, resulting in inhibition of HBV replication [16,17]. Additionally, inhibitors of DNA binding-1 and the X-linked tumor suppressor TSPX interact with proteasome subunit C8 and proteasome regulatory subunit RPN3, respectively, to decrease the stability of HBx [18,19]. A better understanding of the HBx-UPS interaction may provide insights into the mechanisms involved in HBV replication and viral pathogenesis.

p53 is a tumor suppressor that plays a central role in maintaining genomic integrity in response to intracellular and extracellular mutagenic stimuli [20]. It also contributes to host defense against viral infections [21]. The antiviral role of p53 against HBV has been proven through ectopic expression of p53 in a cell culture-based in vitro replication system, which results in inhibition of HBV replication [22]. Further studies have demonstrated direct interactions between p53 and HBx both in vivo and in vitro [23,24]. E6-associated protein (E6AP) was first identified as an E3 ligase that binds to p53, in conjunction with the E6 protein of human papillomavirus types 16 (HPV-16) and 18, to induce its degradation [25,26]. These findings prompted us to investigate the possible role of E6AP in the degradation of HBx by forming a trimeric complex with p53. In this study, we investigated whether the inhibition of HBV replication by p53 was through downregulation of HBx. We also attempted to provide evidence supporting that E6AP is responsible for the p53-dependent ubiquitination and proteasomal degradation of HBx. Next, we attempted to elucidate the detailed mechanism through which E6AP induces HBx ubiquitination. Finally, we attempted to show that p53 inhibits HBV replication via the E6AP-mediated proteasomal degradation of HBx in cell culture systems.

## 2. Materials and Methods

### 2.1. Plasmids

The plasmid pCMV-3 × HA1-HBx (HA-HBx) encodes full-length HBx (genotype D) downstream of the three copies of the influenza virus hemagglutinin (HA) [27]. The 1.2-mer WT HBV replicon containing 1.2 units of the HBV genome (genotype D) and its HBx-null counterpart have been described previously [28]. The plasmids pCMVT N-HA-hE6AP encoding human HA-tagged E6AP (amino acids 262–853) and pCH110 encoding the *Escherichia coli* β-galactosidase (β-Gal) gene were purchased from Addgene (Watertown, MA, USA). The plasmid RC210241, encoding the human Na^+^-taurocholate cotransporting polypeptide (NTCP), was obtained from OriGene (Rockville, MD, USA). Scrambled (SC) shRNA, E6AP shRNA, and p53 shRNA plasmids were purchased from Santa Cruz Biotechnology (Santa Cruz, CA, USA). For the mammalian two-hybrid assay, the PCR fragment from pCMVT N-HA-hE6AP encoding E6AP (amino acids 262–853) was cloned in pSG424, in-frame and downstream of Gal4 (amino acids 1–147) [29] to generate G4-E6AP. In addition, the PCR fragment encoding HBx (amino acids 1–154) from pCMV-3 × HA1-HBx was fused upstream of the VP16 activation domain (amino acids 423–490) in pCMV-VP16 [30] to generate HBx-VP16. The reporter G5E1b-luc contained five copies of the GAL4 binding site upstream of a minimal E1b promoter in pGL3 (Promega, Madison, WI, USA), as described previously [30].

### 2.2. Cell Culture and Transfection

The HepG2 and Hep3B cell lines were obtained from the Korean Cell Line Bank (KCLB, Seoul, South Korea). For transient expression, 2 × 10^5^ cells per 60 mm dish were transfected with 2 μg of the appropriate plasmid(s) using the TurboFect transfection reagent (Thermo Fisher Scientific, Waltham, MA, USA), following the manufacturer’s instructions. Two stable cell lines, HepG2-NTCP and Hep3B-NTCP, were established by transfection with RC210241, followed by selection with 500 μg·mL^−1^ G418 sulfate (Sigma-Aldrich, St. Louis, MO, USA). All cells were cultured in Dulbecco’s modified eagle medium (DMEM, WelGENE, Gyeongsan, South Korea) supplemented with 10% fetal bovine serum (FBS; Capricorn Scientific, Ebsdorfergrund, Germany) and 100 μg·mL^−1^ streptomycin (Gibco, Waltham, MA, USA). Cells were treated with cycloheximide (CHX, Sigma-Aldrich), pifithrin-α (PFT-α, Sigma-Aldrich), or MG132 (Sigma-Aldrich) under the indicated conditions.

### 2.3. HBV Cell Culture Systems

For HBV stock preparation, Hep3B-NTCP cells were transiently or stably transfected with the 1.2-mer HBV replicon plasmid as described above. The culture supernatant was collected for preparation of HBV stocks to determine HBV titers, as described in the next section. HBV infection was conducted in 6-well plates at a multiplicity of genome equivalents (GEQ) of 50 for 4 days, unless otherwise stated. Briefly, 2 × 10^5^ cells were inoculated with 1 × 10^7^ GEQ of HBV and incubated for 24 h in DMEM containing 4% polyethylene glycol 8000 (PEG 8000, Sigma-Aldrich) and 2% dimethyl sulfoxide (DMSO, Sigma-Aldrich). After washing twice with serum-free DMEM, the cells were incubated in DMEM supplemented with 3% FBS, 4% PEG 8000, and 2% DMSO for three days. The culture medium was changed every 3 days, if necessary.

### 2.4. Quantitative Real-Time PCR of HBV DNA

The extracellular HBV titers were measured by quantitative real-time PCR (qPCR) as described previously [31]. Briefly, HBV genomic DNA was purified from the culture supernatant using the QIAamp DNA Mini Kit (Qiagen, Hilden, Germany). For conventional PCR analysis of HBV DNA, genomic DNA was amplified using 2× Τaq PCR Master mix 1 (BioFACT, Daejeon, Korea) and a primer pair, HBV 1399F (5′-TGG TAC CTG CGC GGG ACG TCC TT-3′) and HBV 1632R (5′-AGC TAG CGT TCA CGG TGG TCT CC-3′), as described previously [24]. Additionally, qPCR for HBV was performed as previously described [31]. Briefly, HBV DNA was amplified using SYBR premix Ex Taq II (Takara Bio, Kusatsu, Japan) and a primer pair, HBV 379F (5′-GTG TCT GCG TTT TAT CA-3′) and HBV 476R (5′-GAC AAA CGG GCA ACA TAC CTT-3′), using a Rotor Gene qPCR machine (Qiagen).

### 2.5. Southern Blot Analysis of Intracellular HBV DNA

The Hirt protein-free DNA extraction procedure was used to isolate HBV DNA from HBV-infected cells [32]. Intracellular HBV DNA was detected by Southern blot analysis using the protocol described by Wang et al. [33] with slight modifications. Initially, HBx DNA fragments were obtained by PCR amplification from the 1.2-mer-WT HBV replicon using the primer pair of HBV1399F and HBV 1632R. The amplified DNA fragments were labelled using the DIG Probe Synthesis Kit (Roche, Basel, Switzerland). Hirt-extracted DNA was electrophoresed on a 1.2% agarose gel and transferred onto a nitrocellulose blotting membrane (GE Healthcare Life Sciences, Buckinghamshire, UK). Pre-hybridization was performed for 30 min at 60 °C in 10 mL of DIG Easy Hyb buffer (Roche) followed by overnight hybridization at 54 °C in 3.5 mL of pre-warmed DIG Easy Hyb buffer (Roche) containing 1 µg of DIG-labeled HBx probe. The membrane was then washed with a wash and block buffer (Roche). Probe-target hybrids were detected by spraying an anti-digoxigenin-AP conjugate (Roche) and NBT/BCIP solution (Roche).

### 2.6. HBV e Antigen Enzyme-Linked Immunosorbent Assay

For quantitative analysis of secreted HBV e antigen (HBeAg), 30 μL of culture supernatant was loaded onto 96-well plates using an enzyme-linked immunosorbent assay (ELISA) kit designed for detection of HBeAg, following the manufacturer’s instructions (Cusabio, Houston, TX, USA). The amount of HBeAg was determined using a microplate reader by measuring the optical density of each well at 450 nm (Bio-Rad, Hercules, CA, USA).

### 2.7. Immunofluorescence Analysis

Double-label indirect immunofluorescence assay (IFA) was performed as previously described [34]. Briefly, cells grown on coverslips were fixed in 4% formaldehyde at 20 °C for 15 min and permeabilized in methanol at −20 °C for 10 min. The coverslips were incubated with an appropriate polyclonal antibody and a monoclonal antibody overnight at 4 °C and subsequently incubated with anti-rabbit IgG-rhodamine (Invitrogen Cat No. 31670, Waltham, MA, USA; 1:200 dilution) and anti-mouse IgG-FITC (Sigma-Aldrich Cat No. F0257-1ML; 1:100 dilution) at room temperature for 1 h. Slides were prepared with UltraCruz mounting medium (Santa Cruz Biotechnology) and visualized using an Eclipse fluorescence microscope (Nikon, Tokyo, Japan). Densitometric analysis of the immunofluorescence signal was performed using the ImageJ software (NIH, Bethesda, MD, USA).

### 2.8. Luciferase Reporter Assay

Approximately 1 × 10^5^ cells per well in 12-well plates were transfected with 0.2 μg of a reporter plasmid along with the experimental plasmids under the indicated conditions. To control for the variation in transfection efficiency, 0.1 μg pCH110 was co-transfected as an internal control. Luciferase assay was performed 48 h after transfection using the Luciferase Reporter 1000 Assay System (Promega). Luciferase activity was measured using a microplate luminometer (LuBi; MicroDigital, Seongnam, Korea). β-gal activity was measured using a β-gal assay kit (Thermo Fisher Scientific). Luciferase activity was normalized to the β-gal activity measured in the corresponding cell extracts.

### 2.9. Western Blot Analysis

Cells were lysed in buffer (50 mM Tris-HCl, pH 7.5, 150 mM NaCl, 0.1% SDS, and 1% NP-40) supplemented with protease inhibitors (Roche). The protein concentrations of the cell extracts were measured using a protein assay kit (Bio-Rad). Cell extracts were separated by sodium dodecyl sulfate-polyacrylamide gel electrophoresis and transferred onto nitrocellulose blotting membranes (GE Healthcare Life Sciences). The membranes were incubated with the respective antibodies—anti-p21 (Santa Cruz Biotechnology, Cat No. sc-6246; 1:200 dilution), anti-p53 (Santa Cruz Biotechnology, Cat. sc-126; 1:1000 dilution), anti-p53 upregulated modulator of apoptosis (PUMA, Cell Signaling Cat No. 49765, Danvers, MA, USA; 1:1000 dilution), anti-E6AP (Thermo Scientific Cat No. PA3-843; 1:2000 dilution), anti-HA (Santa Cruz Biotechnology, Cat No. sc-7392; 1:500 dilution), anti-hepatitis B core antigen (HBcAg; Santa Cruz Biotechnology, Cat No. sc-52406; 1:400 dilution), anti-hepatitis B surface antigen (HBsAg; Santa Cruz Biotechnology; Cat No. sc-53300; 1:500 dilution), anti-NTCP (Santa Cruz Biotechnology Cat No. sc-518115; 1:500 dilution), anti-Siah-1 (Abcam, Cat No. ab2237, Cambridge, UK; 1:2000 dilution), anti-Ub (Santa Cruz Biotechnology, Cat No. sc-9133; 1:500 dilution), anti-γ-tubulin (Santa Cruz Biotechnology, Cat. sc-17787; 1:500 dilution), or anti-HBx (Millipore Cat No. MAB8419, Burlington, MA, USA; 1:500 dilution)—and subsequently with a horseradish peroxidase-conjugated anti-mouse secondary antibody (Bio-Rad, Cat No. BR170-6516; 1:3000 dilution), anti-rabbit IgG (H + L)-HRP (Bio-Rad, Cat No. BR170-6515; 1:3000 dilution), or anti-goat IgG (H + L)-HRP (Thermo Scientific Cat No. 31400; 1:10,000 dilution). Protein bands were detected using an ECL kit (Advansta, San Jose, CA, USA) and ChemiDoc XRS imaging system (Bio-Rad).

### 2.10. Immunoprecipitation

An immunoprecipitation (IP) assay was performed using a classic magnetic IP/Co-IP assay kit (Thermo Scientific) according to the manufacturer’s specifications. Briefly, 1 × 10^6^ cells per 100 mm diameter plate were transiently transfected with the indicated amounts of eukaryotic expression plasmids encoding HBx, E6AP, p53, or Ub for 48 h. Whole cell lysates (900 µg) were incubated with anti-HBx antibody (Millipore Cat No. MAB8419; Burlington, MA, USA; 2 µg/mL) overnight at 4 °C to allow formation of immune complexes. After intensive washing, the immune complexes were harvested with Protein A/G magnetic beads (0.25 mg) by incubation for 1 h while mixing. The beads were then collected using a magnetic stand (Pierce, Waltham, MA, USA), and the eluted antigen/antibody complexes were subjected to Western blotting using an anti-HA antibody.

### 2.11. Statistical Analysis

Values indicate the mean ± standard deviation (SD) from at least three independent experiments. A two-tailed Student’s *t*-test was used for all statistical analyses. Differences were considered statistically significant at *p* ≤ 0.05.

## 3. Results

### 3.1. p53 Inhibits HBV Replication in Human Hepatoma Cells

Two human liver cancer cell lines, HepG2 and Hep3B, are frequently used as experimental models for in vitro studies on HBV-related molecular mechanisms [27,35,36]. HepG2 cells, but not Hep3B cells, express wild-type (WT) p53, which provides a unique platform for parallel comparisons of the roles of p53 [35,37]. Therefore, we investigated the effect of p53 on HBV replication by comparing viral replication in these two cell lines. For this purpose, we employed an in vitro HBV infection system using HBV particles derived from a 1.2-mer HBV replicon [28,38]. Western blot analysis of viral proteins in the cell lysates (Figure 1A,B), and measurement of virus particles in the culture medium by qPCR (Figure 1C,D) and HBeAg by ELISA (Figure 1E,F) confirmed the dose- and time-dependent replication of HBV in HepG2-NTCP and Hep3B-NTCP cells, both of which stably express the HBV receptor NTCP [38,39]. The infected cells synthesized three different forms of HBsAg, that is, large (L)-, middle (M)-, and small (S)-HBsAg, each of which was composed of polypeptides of diverse sizes (Figure 1A,B), due to differences in glycosylation patterns of the S and PreS2 domains [40]. Additionally, Western blotting detected small amounts of HBsAg derived from the integrated HBV genome [37,41] in the cell lysates of uninfected Hep3B-NTCP cells (Figure 1A,B), whereas both qPCR and ELISA failed to detect evidence of HBV replication in these cells (Figure 1C–F). Both the dose- and time-dependent infection experiments clearly showed that the levels of intracellular HBV proteins, including HBx and three different forms of HBsAg, were higher in Hep3B-NTCP cells than in HepG2-NTCP cells (Figure 1A,B). In addition, the levels of extracellular virions and HBeAg released from Hep3B-NTCP cells were significantly higher than those released from HepG2-NTCP cells (Figure 1C–F). These results suggest that p53 inhibits HBV replication in human hepatoma cells.

Besides p53 status, HepG2 and Hep3B cells exhibit several other differences, including ethnic origins, distinct chromosome aberrations, HBV DNA integration, and tumorigenicity [35,37], which may affect HBV replication in these two cell lines. This was addressed by measuring HBV replication in HepG2-NTCP cells after p53 knockdown and in Hep3B-NTCP cells after ectopic p53 expression. p53 knockdown stimulated HBV replication in HepG2-NTCP cells, as demonstrated by the elevated levels of intracellular HBV proteins and cccDNA in the cell lysates, and extracellular HBV virions and HBeAg in the culture supernatants (Figure 2A–D). Additionally, ectopic p53 expression inhibited HBV replication in Hep3B-NTCP cells, as evidenced by the lowered levels of intracellular HBV proteins and cccDNA in the cell lysates, and extracellular virions and HBeAg in the culture supernatants (Figure 2A–D). Ectopic p53 expression inhibited HBV replication in HepG2-NTCP cells (Figure 3A,B). Taken together, we concluded that p53 inhibits HBV replication in human hepatoma cells.

### 3.2. p53 Transcriptional Activity Is Not Essential for the Inhibition of HBV Replication

As demonstrated in previous reports, HBV infection upregulated p53 in HepG2-NTCP cells (Figure 1A,B) [17,36]. HBV infection also upregulated two representative p53 targets, p21 and PUMA, in HepG2-NTCP cells (Figure 3A). Ectopic p53 expression further elevated p21 and PUMA levels in HBV-infected HepG2-NTCP cells (Figure 3A). Therefore, it can be hypothesized that p53 inhibits HBV replication through the transcriptional activation of genes involved in cell cycle arrest and apoptotic cell death, such as p21 and PUMA, respectively. To test this hypothesis, we employed a specific p53 inhibitor, pifithrin-α (PFT-α), which prevents the transactivation of p53-responsive genes and thus suppresses p53-dependent apoptosis [42]. Treatment with PFT-α resulted in a dramatic decrease in the levels of p21 and PUMA in HepG2-NTCP cells, irrespective of whether they were infected with HBV or not, and thus, almost completely abolished the effects of HBV infection and ectopic p53 expression on p21 and PUMA levels (Figure 3A). Treatment with PFT-α unexpectedly decreased the levels of p21 and PUMA in the absence of p53 (Figure 3B), which has been demonstrated in a previous report [43]. Additionally, PFT-α almost completely abolished the potential of ectopic p53 expression to upregulate p21 and PUMA in Hep3B-NTCP cells (Figure 3C). These results indicated that PFT-α effectively inhibited p53 transcriptional activity under our experimental conditions.

As demonstrated by the elevated levels of intracellular HBx and HBsAg in the cell lysates and extracellular HBeAg in the culture supernatants (Figure 3A,B), treatment with PFT-α resulted in an increase in HBV replication in HepG2-NTCP cells both in the presence and absence of exogenous p53. Similarly, PFT-α stimulated HBV replication in Hep3B-NTCP cells in the presence of exogenous p53, but not in the absence of p53 (Figure 3C,D). These results indicated that p53 transcriptional activity plays a role in the inhibition of HBV replication in human hepatoma cells. Interestingly, ectopic p53 expression effectively inhibited HBV replication in both HepG2-NTCP and Hep3B-NTCP cells in the presence of PFT-α, as demonstrated by the lowered levels of intracellular HBx and HBsAg in the cell lysates and extracellular HBeAg in the culture supernatants (Figure 3A–D). These results indicate that p53 transcriptional activity is implicated in, but not essential for, inhibition of HBV replication in human hepatoma cells.

### 3.3. p53 Prevents HBx from Stimulating HBV Replication in Human Hepatoma Cells

Next, we investigated the mechanism of p53-mediatied inhibition of HBV replication in human hepatoma cells. We first compared the replication of WT HBV and HBx-null HBV in HepG2-NTCP and Hep3B-NTCP cells to assess the possible role of HBx in p53-mediated inhibition of HBV replication. Consistent with previous reports demonstrating the role of HBx as a positive regulator of HBV replication [7,8], infection with WT HBV produced higher levels of intracellular HBcAg, M-HBsAg, and extracellular viral particles in both cell lines (Figure 4A,B). Interestingly, the fold difference in levels of extracellular virus particle between WT HBV-transfected and HBx-null HBV-transfected Hep3B-NTCP cells (55.5-fold) was over five times higher than that in HepG2-NTCP cells (9.9-fold) (Figure 4B). In addition, the fold difference of viral particles between WT HBV-transfected HepG2-NTCP and Hep3B-NTCP cells (10.1-fold) was over five times higher than that between the two HBx-null HBV-transfected cells (1.8-fold). These results suggest that p53 inhibits HBV replication by preventing HBx from stimulating HBV replication in human hepatoma cells.

To prove the possible role of HBx in p53-mediated inhibition of HBV replication, we attempted to supplement HBx through ectopic expression during HBx-null HBV infection in HepG2-NTCP and Hep3B-NTCP cells. Ectopic HBx expression stimulated the replication of HBx-null HBV in both HepG2-NTCP and Hep3B-NTCP cells, as evidenced by an increase in the levels of intracellular viral proteins and extracellular virus particles. This effect was more dramatic in Hep3B-NTCP cells (Figure 4C,D). Ectopic HBx expression induced a 23.9-fold increase in the level of extracellular HBx-null HBV particles in Hep3B-NTCP cells. This effect was only approximately 4.9-fold in HepG2-NTCP cells (Figure 4D). Additionally, the potential of p53 to inhibit HBx-null HBV replication was much higher in the presence of ectopic HBx, as demonstrated by the fold difference in the levels of HBx-null HBV particles between Hep3B-NTCP cells and HepG2-NTCP in the presence and absence of ectopic HBx (8.5-fold and 1.7-fold, respectively). Therefore, we conclude that HBx plays a role in the p53-mediated inhibition of HBV replication in human hepatoma cells.

### 3.4. p53 Downregulates HBx to Inhibit HBV Replication in Human Hepatoma Cells

To explore the possible mechanism by which p53 prevents HBx from stimulating HBV replication, we examined whether p53 affects HBx levels in human hepatoma cells. Higher levels of HBx were consistently detected during HBV infection in Hep3B-NTCP cells than in HepG2-NTCP cells under all infection conditions (Figure 1A,B). Additionally, p53 knockdown upregulated HBx during HBV infection in HepG2-NTCP cells, while ectopic p53 expression downregulated HBx during HBV infection in Hep3B-NTCP cells (Figure 2A), suggesting that p53 downregulates HBx to inhibit HBV replication in human hepatoma cells. However, it is possible to argue that downregulation of HBx is not a cause, but a result of p53-mediated inhibition of HBV replication. To address this, we examined whether p53 downregulated HBx expression in human hepatoma cells. Ectopic HBx expression consistently resulted in higher levels of HBx in Hep3B cells than in HepG2 cells (Figure 5A). According to data from IFA, a stronger HBx signal was primarily detected in the cytoplasm of Hep3B cells than those of HepG2 cells, where p53 was detected in the nucleus (Figure 5B). In addition, p53 knockdown upregulated HBx in HepG2 cells, while ectopic p53 expression downregulated HBx in Hep3B cells (Figure 5C). Ectopic p53 expression also weakened the HBx signal in the cytoplasm of Hep3B cells (Figure 5D). Based on these observations, we conclude that p53 downregulates HBx in human hepatoma cells.

### 3.5. p53 Induces Ubiquitination and Proteasomal Degradation of HBx

Previous reports have demonstrated that HBx levels are primarily regulated by the UPS [17,44]. Therefore, we first examined whether p53 decreased the stability of HBx in human hepatoma cells. Hep3B cells expressing HBx, with or without exogenous p53, were treated with CHX to block further protein synthesis, while increasing the levels of HBx, p53, and γ-tubulin in these cells (Figure 6A). HBx is a fairly stable protein in Hep3B cells, with a half-life (t_1/2_) of 115.1 min. Ectopic p53 expression reduced the t_1/2_ value of HBx to 63.2 min, indicating that p53 destabilizes HBx in human hepatoma cells. Treatment with PFT-α did not lead to a detectable change in HBx stability in the absence of p53, but resulted in an 11.8% increase in HBx stability in the presence of p53 (Figure 3A), suggesting that p53 transcriptional activity is involved, at least in part, in the destabilization of HBx. These results were consistent with the HBx levels detected during HBV infection in Hep3B-NTCP cells in the presence and absence of PFT-α (Figure 3C). Interestingly, p53 induced a 38.8% reduction in the t_1/2_ value of HBx in the presence of PFT-α, which was slightly lower than the effect (45.1% reduction) obtained in the absence of PFT-α (Figure 6A), indicating that p53 transcriptional activity is not required for the HBx destabilization process. Based on these observations, we conclude that p53 downregulates HBx by decreasing its protein stability through two different mechanisms—one dependent on the p53 transcriptional activity, and one independent of it.

Having established that p53 decreases the stability of HBx, we investigated whether p53 induces ubiquitination of HBx in human hepatoma cells. For this purpose, we introduced HBx and HA-tagged Ub into Hep3B cells with and without p53 and immunoprecipitated HBx in cell lysates. According to the results of the co-immunoprecipitation, HBx was ubiquitinated in Hep3B cells in the absence of p53, as demonstrated by a major band and smeared multiple bands of Ub(n)-HBx (Figure 6B, lane 2). Ectopic p53 expression increased the levels of ubiquitinated HBx, resulting in a decrease in HBx levels in cell lysates (Figure 6B,C, lane 3), indicating that p53 induces ubiquitination and proteasomal degradation of HBx in human hepatoma cells. PFT-α treatment had a minimal effect on HBx ubiquitination in the presence and absence of p53 (Figure 6B,C, lanes 2, 3, 5, and 6). Therefore, ectopic p53 expression could induce ubiquitination of HBx in the presence of PFT-α (Figure 6B,C, lanes 5 and 6), indicating that p53 transcriptional activity is not essential for ubiquitination of HBx in human hepatoma cells.

To further explore the mechanism responsible for p53-induced ubiquitination of HBx, we first compared the binding affinities of two E3 ligases, Siah-1 and E6AP, to HBx in the presence and absence of p53 in Hep3B cells. Ectopic p53 expression upregulated Siah-1 in Hep3B cells expressing HBx in the absence of PFT-α but not in the presence of PFT-α (Figure 4B, lanes 3 and 6), indicating that p53 transcriptionally activates Siah-1 expression in the presence of HBx. This has also been demonstrated in a previous report [17]. In contrast, ectopic p53 expression downregulated E6AP in the absence of PFT-α, however the effect was largely diminished in the presence of PFT-α (Figure 6B,C, lanes 3 and 6), indicating that the ability of p53 to downregulate E6AP in the presence of HBx also depends on the transcriptional activity of p53. According to the co-IP data, ectopic p53 expression increased the amount of Siah-1 and E6AP bound to HBx (Figure 6B,C, lane 3), suggesting that both Siah-1 and E6AP contribute to the p53-induced ubiquitination of HBx. Considering the lower E6AP levels in the cell lysates, increased E6AP binding to HBx in the presence of p53 suggests a possible role for p53 in the interaction between E6AP and HBx. Treatment with PFT-α almost completely abolished the potential of p53 to upregulate Siah-1 in the cell lysates and the amount of Siah-1 bound to HBx (Figure 6B,C, lane 6), indicating that Siah-1-mediated HBx ubiquitination depends on p53 transcriptional activity. In contrast, treatment with PFT-α upregulated both E6AP in the cell lysates and the amount of E6AP bound to HBx in the presence of p53 (Figure 6B,C, lane 6), suggesting that the interaction between E6AP and HBx does not require p53 transcriptional activity. Taken together, we conclude that although both E6AP and Siah-1 are implicated in the p53-induced ubiquitination of HBx, E6AP plays a more critical role during ubiquitination of HBx through a mechanism that does not require p53 transcriptional activity.

### 3.6. E6AP Is Responsible for the p53-Induced Ubiquitination of HBx

Since it has been established that p53 activates Siah-1 expression at the transcriptional level to induce ubiquitination of HBx [16,17], we focused on the p53-dependent E6AP-mediated downregulation of HBx. To confirm that E6AP is responsible for p53-induced downregulation of HBx, we examined whether E6AP overexpression or knockdown affects HBx levels in the presence and absence of p53. Ectopic E6AP expression downregulated HBx, while E6AP knockdown upregulated HBx in HepG2 cells (Figure 7A). In contrast, neither ectopic E6AP expression nor E6AP knockdown induced recognizable changes in HBx levels in Hep3B cells (Figure 7B). IFA data also clearly showed that ectopic E6AP expression downregulated HBx in the presence of p53, but not in the absence of p53 (Figure 7C). The Pearson correlation coefficients showed an inverse correlation between HBx and E6AP in HepG2 cells, but not in Hep3B cells (Figure 7D). Additionally, both ectopic E6AP expression and E6AP knockdown affected HBx levels in Hep3B cells in the presence of p53 (Figure 7E), similar to that in HepG2 cells (Figure 7A). Therefore, we conclude that E6AP is responsible for the p53-induced downregulation of HBx in human hepatoma cells.

To examine whether E6AP decreases the stability of HBx protein in a p53-dependent manner, we treated HepG2 and Hep3B cells expressing HBx with or without exogenous E6AP with CHX and analyzed the HBx and γ-tubulin levels in these cells. The t_1/2_ value of HBx in HepG2 and Hep3B cells was 78.1 min and 138.6 min, respectively (Figure 8A), which was consistent with HBx levels in these cell lines (Figure 1A). Ectopic E6AP expression resulted in a 56.8% decrease in HBx stability in HepG2 cells (t_1/2_ = 33.7 min), while it caused a 3.2% decrease in Hep3B cells (t_1/2_ = 134.1 min). In contrast, ectopic E6AP expression led to a 76.2% decrease in HBx stability in Hep3B cells expressing exogenous p53 (t_1/2_ = 32.9 min), suggesting that E6AP decreased the stability of HBx protein in a p53-dependent manner.

### 3.7. p53 and E6AP Help Each Other in Binding to HBx in Human Hepatoma Cells

We investigated how E6AP induced p53-dependent HBx ubiquitination. In the presence of the proteasome inhibitor MG132, ectopic E6AP expression minimally affected total p53 and HBx levels in HepG2 and Hep3B cells (Figure 8B). Ectopic E6AP expression substantially increased the amount of E6AP bound to HBx in HepG2 cells, resulting in a dramatic increase in HBx ubiquitination (Figure 8B, Lane 3). However, ectopic E6AP expression barely managed to add a small amount of E6AP to HBx in Hep3B cells, resulting in minimal effect on HBx ubiquitination (Figure 8B, lane 6). Ectopic p53 expression facilitated the binding of E6AP to HBx in Hep3B cells, resulting in a dramatic increase in HBx ubiquitination (Figure 8B,C). Interestingly, ectopic E6AP expression also increased the amount of p53 bound to HBx in HepG2 cells (Figure 8B, lane 3), suggesting that p53 and E6AP help each other bind to HBx in human hepatoma cells.

The role of p53 in the binding of E6AP to HBx was further investigated using a mammalian two-hybrid assay with the reporter plasmid G5E1b-luc, which contains five copies of GAL4 binding site [30]. High reporter gene activity from G5E1b-luc was observed when both G4-E6AP and HBx-VP16 were included in the HepG2 cells because of the interaction between E6AP and HBx (Figure 8D, column 3). The interaction between E6AP and HBx in the absence of p53 also induced a significantly higher reporter activity from G5E1b-luc in Hep3B cells, although the strength was much lower than that in HepG2 cells (Figure 8D, columns 3 and 8). Consistent with the data obtained from the co-immunoprecipitation (Figure 8B,C), p53 knockdown decreased the luciferase activity in HepG2 cells, while ectopic p53 expression increased the luciferase activity in Hep3B cells (Figure 8D), suggesting that E6AP and HBx can interact with a much higher affinity with the aid of p53.

### 3.8. p53 Inhibits HBV Replication by Downregulating HBx via E6AP-Mediated Proteasomal Degradation

Finally, we investigated whether p53 inhibits HBV replication via the E6AP-mediated downregulation of HBx. Ectopic E6AP expression downregulated HBx during HBV infection, resulting in the inhibition of viral replication, as demonstrated by a decrease in intracellular HBcAg and HBsAg levels and extracellular HBV particles in HepG2-NTCP cells, whereas none of these effects were observed in Hep3B-NTCP cells (Figure 9A,B). Additionally, ectopic p53 expression was sufficient to restore the ability of E6AP to downregulate HBx and inhibit HBV replication in Hep3B cells (Figure 9A,B). Moreover, E6AP minimally affected the replication of HBx-null HBV in HepG2-NTCP cells, as demonstrated by the effects of ectopic E6AP expression on the levels of intracellular viral proteins and extracellular HBV particles (Figure 9C,D). Taken together, we concluded that p53 inhibits HBV replication by downregulating HBx via E6AP-mediated ubiquitination and proteasomal degradation in human hepatoma cells.

## 4. Discussion

p53 plays a central role in maintaining genomic integrity in response to intracellular and extracellular mutagenic stimuli [20]. The role of p53 as the guardian of the genome contributes to the host defense system against viral infections [21]. For example, p53 often induces G_1_ arrest in response to viral infection to inactivate cellular DNA synthesis machinery, causing abortive infection. Thus, several DNA viruses have evolved various strategies to inactivate or degrade p53 for genome replication, providing a potent oncogenic mechanism for DNA tumor viruses [45]. p53-mediated apoptosis also provides a powerful means of preventing the spread of infectious viruses from infected cells [46]. p53 can also directly induce antiviral immune responses by activating the expression of innate immunity-related genes, such as IRF9, TRL3, ISG15, and MCP-1 [47,48,49,50]. Moreover, p53 may directly inhibit HBV replication by inhibiting transcription from the HBV core promoter and enhancers [51,52]. Therefore, it was not surprising to observe that absence of p53 results in increased levels of HBV replication in HBV infection, as compared to its levels in the presence of p53 (Figure 1). p53 knockdown stimulated HBV replication in HepG2-NTCP cells, while ectopic p53 expression inhibited HBV replication in Hep3B-NTCP cells (Figure 2). Treatment with PFT-α, a potent p53 inhibitor, stimulated HBV replication in the presence of p53 (Figure 3), suggesting that p53 can inhibit HBV replication, at least in part, by transcriptionally activating the expression of target genes involved in the regulation of the cell cycle, apoptosis, and innate immunity, or directly inactivating the HBV core promoter and enhancers. Interestingly, the potential of p53 to inhibit HBV replication remained active after treatment with PFT-α, which completely inhibited the ability of p53 to activate the expression of its target genes, such as p21 and PUMA (Figure 3), suggesting that p53 can act as a negative regulator of HBV replication via another mechanism that does not involve its transcriptional activity.

Considering the essential role of HBx during HBV replication both in vivo and in vitro [5,6,7,8,53], HBx can be considered as a preferential target of p53 to inhibit HBV replication. The role of HBx as a positive regulator of HBV replication was more evident in the absence of p53 (Figure 4). Additionally, the ability of p53 to inhibit replication of HBx-null HBV was relatively lower than its ability to inhibit the replication of WT HBV (Figure 4). These results suggest that p53 inhibits HBV replication by restricting the potential of HBx as a positive regulator of HBV replication. As p53 is still able to inhibit the replication of HBx-null HBV (Figure 4), other viral proteins may be involved in the p53-mediated anti-HBV defense. It is also possible to assume that p53 inhibits the replication of HBx-null HBV by directly inactivating the HBV core promoter and enhancer, as previously described [51,52].

Transient expression of HBx consistently produced higher protein levels in the absence of p53 (Figure 5), as previously demonstrated [54]. Therefore, it was possible to speculate that p53 inhibits HBV replication by downregulating HBx. Consistent with the regulatory function of HBx, it is usually present at low levels in HBV-infected cells because it is rapidly degraded by the UPS [14,15]. The role of p53 as a negative regulator of HBx stability has been proposed based on data from both p53 overexpression and knockdown experiments in HCC cell lines [54], which was also demonstrated in this study (Figure 6). It has been further demonstrated that Siah-1, an E3 ligase, induces ubiquitination and proteasomal degradation of HBx [16]. As p53 activates the expression of Siah-1 at the transcriptional level [17], Siah-1 obviously contributes to the role of p53 as a negative regulator of HBx stability. However, the observation that p53 still decreased the protein stability of HBx when PFT-α completely abolished the potential of p53 to activate Siah-1 (Figure 6) suggests the presence of another E3 ligase involved in p53-induced HBx ubiquitination. E6AP, an E3 ligase, was activated by p53 through a direct protein–protein interaction, to induce ubiquitination and proteasomal degradation of HBx, which did not require p53 transcriptional activity (Figure 8), providing a new mechanism of action of p53 as a negative regulator of HBx stability. According to the data from IFA, HBx was primarily detected in the cytoplasm, whereas p53 was mostly localized in the nucleus (Figure 5B), which sets a physically unfavorable condition for p53-HBx interactions. However, considering its roles in the dysregulation of intracellular signaling in the cytoplasm as well as in the regulation of viral and cellular genes in the nucleus, which have been extensively reported so far [4,7,8,11,12,28], it is obvious that HBx is localized both in the cytoplasm and nucleus of HepG2 and Hep3B cells. The HBx level in the nucleus might be too low to make a detectable signal by IFA.

The mechanisms by which Siah-1 and E6AP induce HBx ubiquitination appear to differ. Unlike Siah-1, which can induce HBx ubiquitination independent of p53 [17], E6AP requires p53 to execute its role as an E3 ligase of HBx (Figure 8). Thus, it is likely that the ubiquitination of HBx in the absence of p53 (Figure 5B) was mostly due to the action of Siah-1. According to the data from co-immunoprecipitation, binding of E6AP to HBx was limited in the absence of p53, while it was significantly increased in the presence of p53 (Figure 8B). As E6AP also strengthened the binding affinity of p53 to HBx, it is likely that p53 and E6AP cooperate to bind HBx (Figure 8B). It is unclear whether E6AP binding to p53 potentiates E3 ligase activity via conformational changes or simply enables it to contact HBx as a substrate. It is also unknown why two different E3 ligases are involved in HBx ubiquitination. Thus, E6AP and Siah-1 may play different roles in HBV replication. According to a previous report, HBx actively induces Siah-1 expression via upregulation of p53 to modulate its own protein levels via a negative feedback loop involving p53 and Siah-1 to control HBV propagation [17]. In contrast, this study showed that HBx downregulated E6AP only in the presence of p53 (Figure 7), which may provide a means to evade E6AP-mediated degradation, as demonstrated by HCV core protein [55]. Interestingly, treatment with PFT-α decreased the amount of Siah-1 bound to HBx in the presence of p53 but failed to induce recognizable changes in HBx ubiquitination patterns (Figure 6B,C). This was probably because E6AP, rather than Siah-1, binds to HBx in the presence of p53. Therefore, Siah-1 and E6AP may compensate for each other to ubiquitinate HBx. It would also be interesting to investigate whether Siah-1 and E6AP are differentially involved in the ubiquitination of six Lys residues in HBx [56]. Regardless, as both Siah-1 and E6AP require p53 to effectively induce HBx ubiquitination, it is unclear whether the levels of HBx are higher in p53-negative human hepatoma cells. More extensive studies are required to clarify their exact roles in Ub-dependent proteasomal degradation of HBx during HBV replication.

This study showed that E6AP induced ubiquitination of HBx only in the presence of p53 (Figure 8B). E6AP has been shown to induce ubiquitination of HPV E6 and HCV core proteins in the presence of p53 [34,57]. Therefore, it is likely that p53 has evolved a common mechanism involving E6AP to target HPV E6, HCV core protein, and HBx to provide anti-viral strategies against three human tumor viruses. According to a previous report [22], HBx rescues the inhibitory function of p53 in human hepatoma cells, but the exact mechanism remains unclear. According to the present study, HBx facilitates p53 binding to E6AP to form a trimeric complex of E6AP, HBx, and p53, which may trigger ubiquitination and proteasomal degradation of p53, as demonstrated by HPV E6 [58]. Therefore, it is likely that the HBx to p53 ratio is an important factor in determining the fate of viral replication. The antagonistic interaction between p53 and HBx, which depends on the E3 ligase activity of E6AP, may be critical for their roles as tumor suppressors and viral oncoproteins, respectively, during the course of HBV pathogenesis.

## 5. Conclusions

The tumor suppressor p53 has been implicated in the host defense system against HBV infection, although the underlying mechanism remains poorly understood. This study demonstrates that p53 activates E3 ligase E6AP through direct interaction to induce ubiquitin-dependent proteasomal degradation of HBx, resulting in downregulation of HBx and subsequent inhibition of HBV replication in human hepatoma cells. These findings may provide insights into the regulation of HBx during HBV replication, especially from the perspective of p53 status, which can help us understand viral pathogenesis and tumorigenesis.

## Figures and Tables

**Figure 1 viruses-14-02313-f001:**
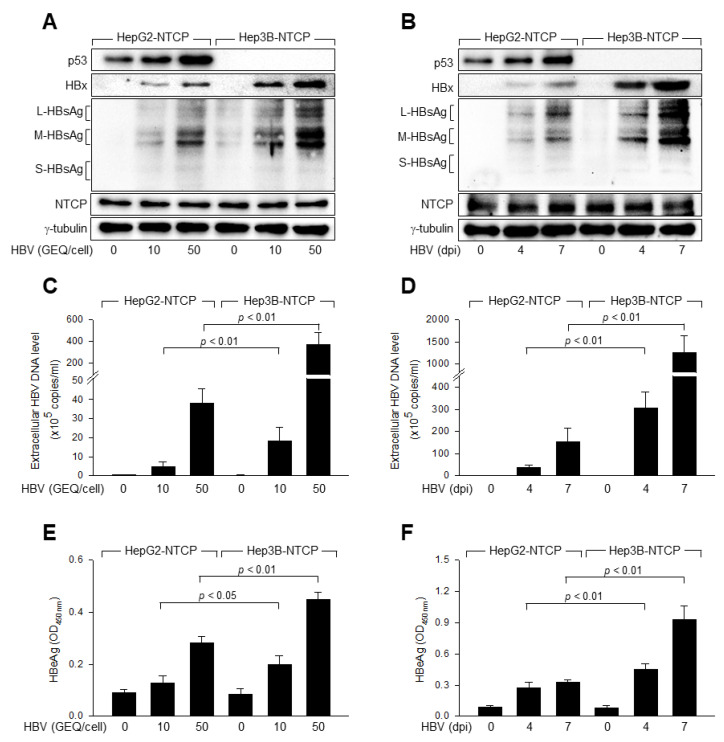
HBV replication is impaired in p53-positive human hepatoma cells. (**A**) HepG2-NTCP and Hep3B-NTCP cells were infected with HBV at the indicated genome equivalent (GEQ) per cell for 4 days. Levels of HBx and three HBV envelope proteins, large hepatitis B surface antigen (L-HBsAg), middle HBsAg (M-HBsAg), and small HBsAg (S-HBsAg) in cell lysates were measured by Western blotting. The levels of p53, Na^+^-taurocholate cotransporting polypeptide (NTCP), and γ-tubulin were also determined. (**B**) HepG2-NTCP and Hep3B-NTCP cells were infected with HBV at 50 GEQ/cell as in (**A**), followed by Western blotting at 0, 4, and 7 days post-infection (dpi). (**C**) Levels of HBV particles in cell supernatants prepared in (**A**) were determined by quantitative real-time PCR (qPCR) (*n* = 7). (**D**) The levels of HBV particles in cell supernatants prepared in (**B**) were determined by qPCR (*n* = 6). (**E**) The levels of HBeAg in cell supernatants prepared in (**A**) were measured by ELISA (*n* = 4). (**F**) The levels of HBeAg in cell supernatants prepared in (**B**) were measured by ELISA (*n* = 4).

**Figure 2 viruses-14-02313-f002:**
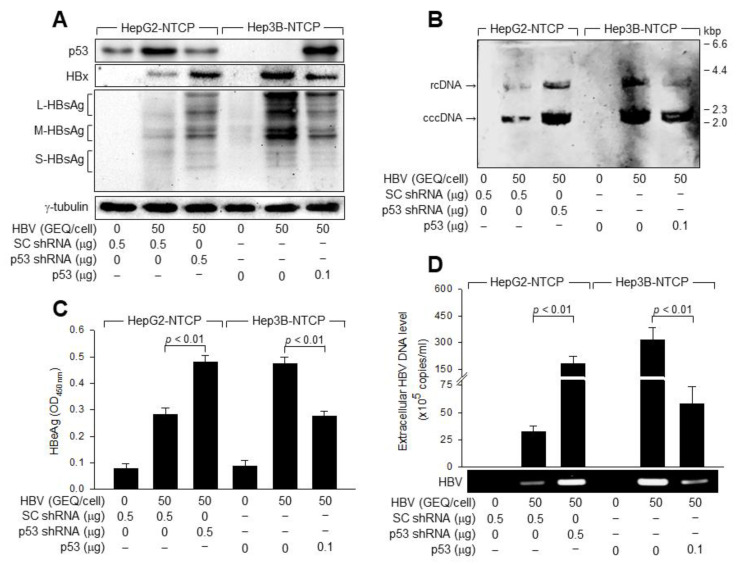
p53 inhibits HBV replication in human hepatoma cells. HepG2-NTCP and Hep3B-NTCP cells were transfected with the indicated amounts of scrambled (SC) shRNA, p53 shRNA, or p53 expression plasmid for 24 h, and then either mock-infected or infected with HBV at 50 GEQ/cell for 4 days. (**A**) Levels of the indicated proteins in cell lysates were measured by Western blotting. (**B**) Southern blotting was performed to detect HBV DNA extracted from the infected cells using the Hirt method. (**C**) The levels of HBeAg released from infected cells were measured by ELISA (*n* = 4). (**D**) Levels of HBV particles released from infected cells were determined by conventional PCR and qPCR (*n* = 6).

**Figure 3 viruses-14-02313-f003:**
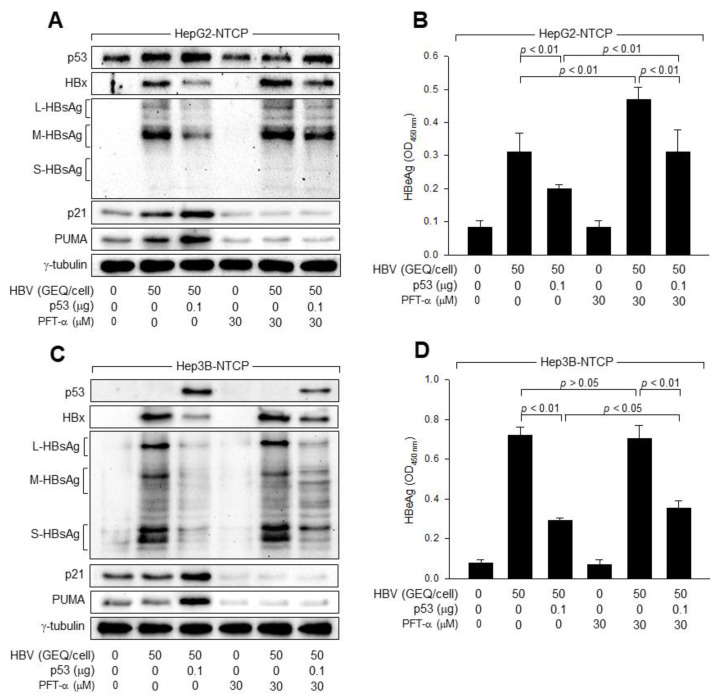
p53 transcriptional activity is not essential for inhibition of HBV replication. (**A**,**B**) HepG2-NTCP and Hep3B-NTCP cells were transfected with either an empty vector or p53 expression plasmid for 24 h and either mock-infected or infected with HBV at 50 GEQ/cell for 4 days in the presence or absence of pifithrin-alpha (PFT-α), followed by Western blotting. (**B**) Levels of HBeAg released from the cells prepared in (**A**) were measured by ELISA (*n* = 4). (**D**) The levels of HBeAg released from the cells prepared in (**C**) were measured by ELISA (*n* = 5).

**Figure 4 viruses-14-02313-f004:**
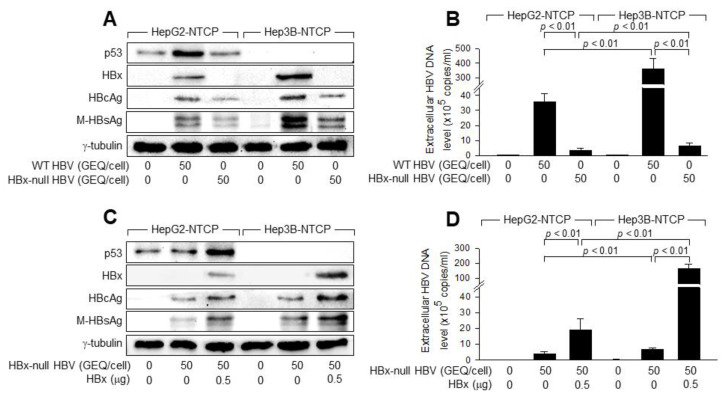
p53 prevents HBx from stimulating HBV replication in human hepatoma cells. (**A**) HepG2-NTCP and Hep3B-NTCP cells were either mock-infected or infected with WT HBV or HBx-null HBV at 50 GEQ/cell for four days, followed by Western blotting. (**B**) Levels of HBV particles released from the cells prepared in (**A**) were determined by qPCR (*n* = 7). (**C**) HepG2-NTCP and Hep3B-NTCP cells were transfected with either an empty vector or HBx expression plasmid for 24 h and either mock-infected or infected with HBx-null HBV at 50 GEQ/cell for four days, followed by Western blotting. (**D**) Levels of HBV particles released from the cells prepared in (**C**) were determined by qPCR (*n* = 5).

**Figure 5 viruses-14-02313-f005:**
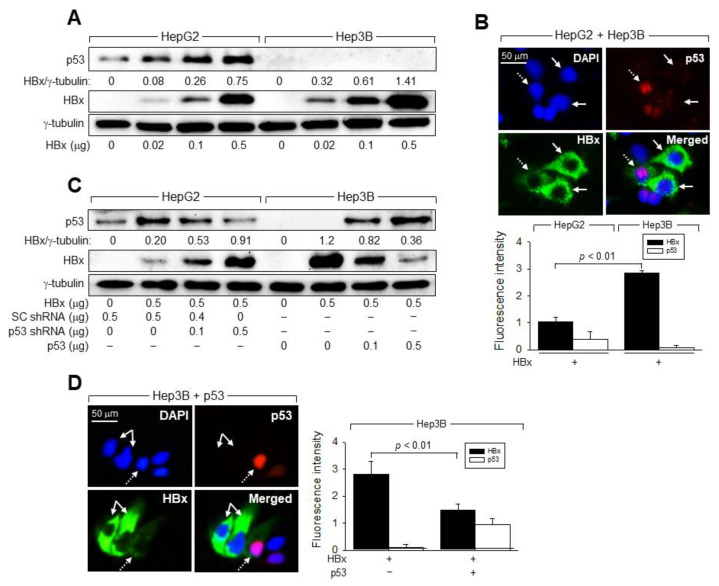
p53 downregulates HBx expression in human hepatoma cells. (**A**) HepG2 and Hep3B cells were transiently transfected with increasing concentrations of HBx expression plasmid for 48 h, followed by Western blotting. Each band was quantified using ImageJ image analysis software (NIH, Bethesda, MD, USA), and the values indicate the levels of HBx relative to the loading control (γ-tubulin). (**B**) An equal number of HepG2 and Hep3B cells grown on coverslips were transiently transfected with HBx expression plasmid for 48 h. Cells were fixed, incubated with anti-HBx monoclonal and anti-p53 polyclonal antibodies, and then incubated with anti-mouse IgG-FITC and anti-rabbit IgG-rhodamine antibodies to visualize HBx (green) and p53 (red), respectively. Nuclei (blue) were stained with 4′, 6-diamidino-2-phenylindole (DAPI). Representative images are shown. Hep3B cells expressing HBx are shown as straight arrows, whereas one HepG2 cell expressing both HBx and p53 is indicated with a broken arrow. The fluorescence intensity was analyzed using ImageJ image analysis software (*n* = 3). Scale bar = 50 μm. (**C**) HepG2 and Hep3B cells were transiently transfected with the HBx expression plasmid along with SC shRNA, p53 shRNA, or p53 expression plasmid for 48 h, followed by Western blotting. (**D**) Hep3B cells were co-transfected with HBx and p53 expression plasmids for 48 h. Cells were subjected to double-label indirect immunofluorescence assays as described in (**B**) to visualize HBx (green) and p53 (red). Hep3B cells expressing HBx are shown with straight arrows, whereas cells expressing both HBx and p53 are indicated by broken arrows. The fluorescence intensity was analyzed using ImageJ image analysis software (*n* = 3). Scale bar = 50 μm.

**Figure 6 viruses-14-02313-f006:**
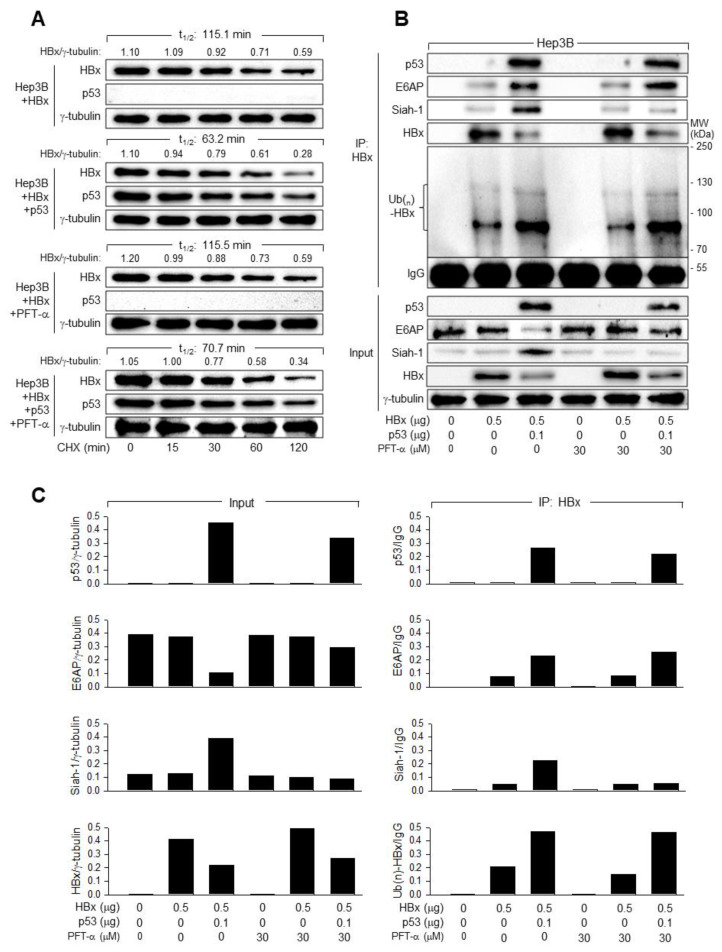
p53 induces ubiquitin-dependent proteasomal degradation of HBx. (**A**) Hep3B cells were transfected with HBx expression plasmid along with either an empty vector or p53 expression plasmids for 48 h in the presence or absence of 30 μM PFT and treated with 50 μM cycloheximide (CHX) for the indicated time before harvesting and Western blotting. The levels of HBx and γ-tubulin were quantified, as described in Figure 5A, to determine the half-life (t_1/2_) of HBx. (**B**) Hep3B cells were transfected with HBx and HA-Ub expression plasmids along with either an empty vector or p53 expression plasmids for 48 h in the presence or absence of 30 μM PFT-α. Total HBx protein in cell lysates was immunoprecipitated with an anti-HBx antibody and subjected to Western blotting using anti-p53, anti-E6AP, anti-Siah-1, anti-HBx, and anti-HA antibodies to detect p53, E6AP, Siah-1, HBx, and HA-Ub-complexed HBx, respectively. The input shows the levels of the indicated proteins in the cell lysates. (**C**) Each band in (**B**) was quantified as described in Figure 5A, and the values show the levels of each protein relative to γ-tubulin in the cell lysates (left) and the levels of each protein relative to IgG in the immunoprecipitates (right).

**Figure 7 viruses-14-02313-f007:**
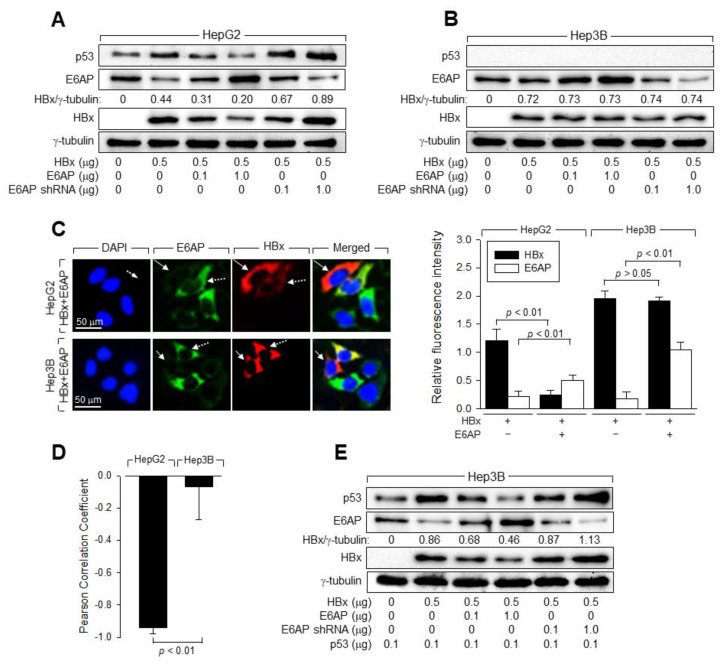
E6AP is responsible for the p53-mediated downregulation of HBx in human hepatoma cells. (**A**,**B**,**E**) HepG2 and Hep3B cells were transiently transfected with the HBx expression plasmid along with the indicated plasmids for 48 h, followed by Western blotting. HBx levels were quantified as described in Figure 5A. (**C**) HepG2 and Hep3B cells grown on coverslips were co-transfected with HBx and E6AP expression plasmids for 48 h and processed for double-label indirect immunofluorescence, as shown in Figure 5B, to visualize E6AP (green) and HBx proteins (red). A representative cell expressing both HBx and exogenous E6AP is indicated by the broken arrow, whereas a cell expressing HBx alone is indicated by the straight arrow. The fluorescence intensity was analyzed using ImageJ image analysis software (*n* = 3). Scale bar = 50 μm. (**D**) Pearson’s correlation coefficient was calculated to demonstrate an inverse correlation between HBx and E6AP in the presence of p53 as in (**A**) or absence of p53 as in (**B**) (*n* = 3).

**Figure 8 viruses-14-02313-f008:**
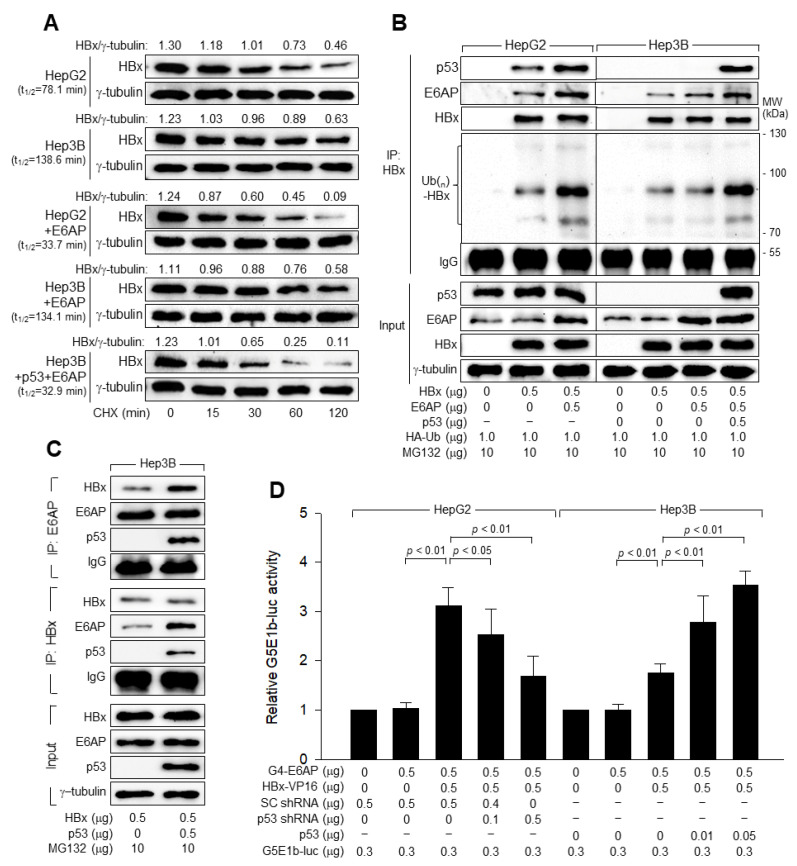
p53 induces E6AP-mediated ubiquitination and proteasomal degradation of HBx. (**A**) HepG2 and Hep3B cells were transfected with HBx, E6AP, and p53 expression plasmids for 48 h as described in (**B**) and treated with 50 μM CHX for the indicated time before harvesting. The levels of HBx and γ-tubulin were quantified, as described in Figure 5A, to determine the t_1/2_ value of HBx. (**B**) HepG2 and Hep3B cells were transfected with HBx expression plasmid along with the indicated plasmids for 48 h and then treated with 10 μM MG132 for 4 h before harvesting. Total HBx protein in the cell lysates was immunoprecipitated with an anti-HBx antibody and subjected to Western blotting using anti-p53, anti-E6AP, anti-HBx, and anti-HA antibodies to detect p53, E6AP, HBx, and HA-Ub-complexed HBx, respectively. The input shows the levels of the indicated proteins in the cell lysates. (**C**) Hep3B cells were transfected with the HBx expression plasmid along with either an empty vector or p53 expression plasmid for 48 h and then treated with 10 μM MG132 for 4 h before harvesting. E6AP and HBx in cell lysates were immunoprecipitated with anti-E6AP and HBx antibodies, respectively. Western blotting was performed to detect HBx, E6AP, and p53 in the immunoprecipitates and cell lysates. (**D**) For mammalian two-hybrid assays, HepG2 and Hep3B cells were transfected with G5E1b-luc, pSG424-E6AP (G4-E6AP), and pCMV HBx-VP16 (HBx-VP16), along with SC shRNA, p53 shRNA, or p53 expression plasmid for 48 h, followed by a luciferase assay. The values indicate the relative luciferase activity compared with the basal level of the control (*n* = 5).

**Figure 9 viruses-14-02313-f009:**
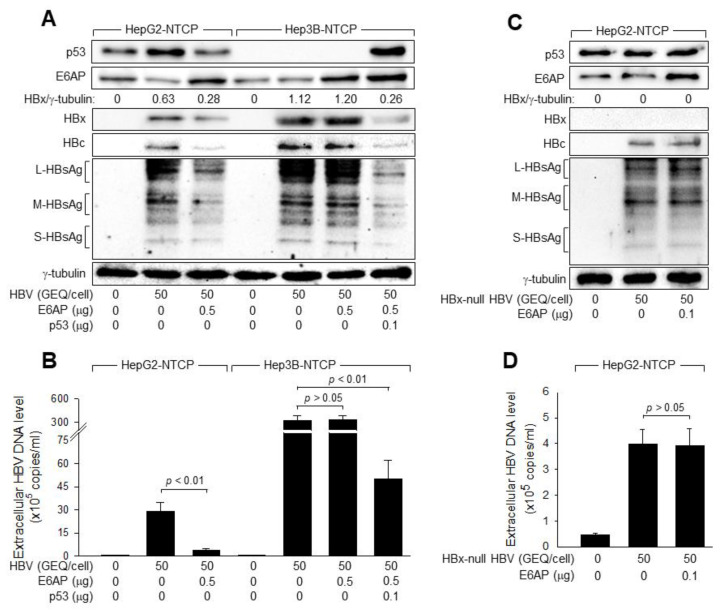
p53 inhibits HBV replication via the E6AP-mediated proteasomal degradation of HBx. (**A**) HepG2-NTCP and Hep3B-NTCP cells were transfected with the indicated amounts of E6AP and p53 expression plasmids for 24 h and either mock-infected or infected with HBV at 50 GEQ/cell for an additional 4 days. (**B**) Levels of HBV particles released from the cells prepared in (**A**) were determined by qPCR (*n* = 5). (**C**) HepG2-NTCP cells were transfected with an empty vector or E6AP expression plasmid for 24 h and either mock-infected or infected with HBx-null HBV at 50 GEQ/cell for an additional four days. (**D**) Levels of HBV particles released from the cells prepared in (**C**) were determined by qPCR (*n* = 5).

## Data Availability

The data presented in this study are available from the corresponding author upon reasonable request.

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
