# Peer review of "Tumor Suppressor p53 Inhibits Hepatitis B Virus Replication by Downregulating HBx via E6AP-Mediated Proteasomal Degradation in Human Hepatocellular Carcinoma Cell Lines"

_viruses, 2022, doi:10.3390/v14102313_

Round 1

Reviewer 1 Report

In this study, Lim et al. investigated the underlying mechanisms of how p53 inhibits HBV replication in human hepatoma cells. They found that p53 can downregulates HBx via ubiquitin-dependent proteasomal degradation by forming p53-E6AP-HBx complex. Furthermore, by using a p53 inhibitor, pifithrin-α treatment, the authors concluded that p53 transcriptional activity for the HBV inhibition effect mentioned above was not essential. This is a well-done study with clearly presented data and the research may provide insights into the regulation mechanisms of p53 on HBV replication through HBx.

Minor comments

1. What’s the LoD for the HBV DNA qPCR quantification system used in this study?

2. Line 106-108: When making the HBV stocks, did the authors concentrate on the collected supernatants? Is the concentration high enough for infection without further concentration?

3. Based on the immunofluorescence data (fig. 5B), a stronger HBx signal was detected in the cytoplasm while p53 was found in the nucleus, so how to understand the forming of p53-E6AP-HBx complex evidenced by the IP experiment data in Fig. 8C.

4. In Figure 8, legends for panel D were lost.

5. It’s reasonable to use the MG132 in Fig 8 to inhibit the ubiquitin depended HBx degradation in order to see the detectable HBx signal. Have the authors also performed MG132 treatment experiments when testing p53 can inhibit HBV replication (for example, similar experiments in Fig.1) to further prove that the inhibition effect is ubiquitin depended HBx degradation?

6. In Figure 9A, how to explain the E6AP decrease upon HBV infection?

Author Response

1. What’s the LoD for the HBV DNA qPCR quantification system used in this study?

Answer:  The limit-of-detection (LoD) of the HBV DNA qPCR used in this study was 40 GEQ/ml with a sample input volume of 2 ml.

2. Line 106-108: When making the HBV stocks, did the authors concentrate on the collected supernatants? Is the concentration high enough for infection without further concentration?

Answer: It was possible to prepare HBV stocks with titers of 1~10 × 108 GEQ/ml. Therefore, 100 ml or less was enough to infect 2 × 105 cells at a multiplicity of GEQ of 50, thus additional concentration step was unnecessary.

3. Based on the immunofluorescence data (fig. 5B), a stronger HBx signal was detected in the cytoplasm while p53 was found in the nucleus, so how to understand the forming of p53-E6AP-HBx complex evidenced by the IP experiment data in Fig. 8C.

Answer: The data in Fig. 5B and D shows, as the reviewer pointed out, that HBx is exclusively present in the cytoplasm. However, it is generally considered that HBx is localized both in the cytoplasm and nucleus to perform its roles in the regulation of intracellular signaling in the cytoplasm and in the regulation of viral and cellular genes in the nucleus, which have been extensively reported. Please refer to the new description in the Discussion section (page 17, lines 818-825).

4. In Figure 8, legends for panel D were lost.

Answer: Sorry. We made a mistake during manuscript transfer from the original manuscript to the Viruses template. It was corrected, as shown in lines 753-758.

5. It’s reasonable to use the MG132 in Fig 8 to inhibit the ubiquitin depended HBx degradation in order to see the detectable HBx signal. Have the authors also performed MG132 treatment experiments when testing p53 can inhibit HBV replication (for example, similar experiments in Fig.1) to further prove that the inhibition effect is ubiquitin depended HBx degradation?

Answer: It might be possible to confirm a direct role of proteasomal degradation in the p53-mediated inhibition of HBV replication, if HBV normally replicates in the presence of MG132. In general, cells are treated with MG132 for a short period (4 h) before harvesting, mainly because of its unstable properties and toxic effects on cell growth. Therefore, we afraid to treat with MG132 for 3-4 days during HBV infection to see its effects on virus replication, which may obviously cause side effects on cell growth and HBV replication.

6. In Figure 9A, how to explain the E6AP decrease upon HBV infection?

Answer: As the reviewer pointed out, HBV infection lowers E6AP levels in HepG2-NTCP but not in Hep3B-NTCP cells, as shown in Fig. 9A. In addition, ectopic HBx expression downregulates E6AP levels in the presence of p53 but not in the absence of p53 (Fig. 7A and B), suggesting that HBx is involved in this process. Please refer to the relevant description in the Discussion section in lines 839-841.

Reviewer 2 Report

Major comments:

1.       From Figure 1, it’s hard to conclude that HBV replication is impaired in p53-positive human hepatoma. Because the authors used 2 different cell lines, and the efficiency of infection and replication of HBV may be different significantly. We could only get the hypothesis. If you want to confirm that p53 impairs the HBV replication, it’s better to use the same cell line with or without p53. Thus, the authors should reorganize the manuscript.

2.       To determine whether p53 Transcriptional Activity is essential for the inhibition of HBV replication, the authors should investigate the function of p53 on HBX promoter activity.

3.       Why did the authors use the HBV infecting NTCP cell lines for all the invitro studies, the author should also use the HBV replicon. We don’t know whether p53 have function on HBV infection.

4.       For some experiments, HBV RNA levels should be detected.

5.       Figure 6, the quantification is needed.

6.       If necessary, human primary hepatocytes could be used.

Author Response

1. From Figure 1, it’s hard to conclude that HBV replication is impaired in p53-positive human hepatoma. Because the authors used 2 different cell lines, and the efficiency of infection and replication of HBV may be different significantly. We could only get the hypothesis. If you want to confirm that p53 impairs the HBV replication, it’s better to use the same cell line with or without p53. Thus, the authors should reorganize the manuscript.

Answer: The reviewer may miss or misunderstand our data and related descriptions, which can give answers to the issues raised by the reviewer. Please find the data in Fig. 2 and read relevant descriptions on page 5 in lines 235-260.

2. To determine whether p53 Transcriptional Activity is essential for the inhibition of HBV replication, the authors should investigate the function of p53 on HBX promoter activity.

Answer: Of course, p53 may inhibit HBV replication by activating transcription of its target genes such as p21 and PUMA or directly inhibiting the HBV promoter and enhancers, as previously described. However, the main finding of the present study is that p53 can inhibit HBV replication in the presence of PFT-a (a potent p53 inhibitor), which means that p53 can act as a negative regulator of HBV replication via another mechanism that does not involve its transcriptional activity. To respond to the reviewer’s comment, we added new descriptions in the Discussion section on page 16.

3. Why did the authors use the HBV infecting NTCP cell lines for all the invitro studies, the author should also use the HBV replicon. We don’t know whether p53 have function on HBV infection.

Answer: The HBV replicon has been used as an in vitro HBV replication system. Recently, however, as the efficiency of HBV replication in HepG2-NTCP cells is improved by treatment with PEG, DMSO, and others, the HBV infection system tends to replace the Replicon system in studying the HBV replication cycle in vitro. According to our preliminary studies, p53 also inhibited replication of HBV derived from the replicon (data not shown), which does not necessarily mean that p53 does not affect the early stages of HBV replication such as viral entry. The present study focused on the effects of p53 on HBx and HBV replication. It is relatively well established that HBx positively regulates HBV replication through the several stages of HBV replication, particularly, HBV transcription. It might be interesting to investigate whether p53 directly or via HBx affects HBV entry into host cells, as the reviewer suggested. However, this is not the main purpose of the present study.   

4. For some experiments, HBV RNA levels should be detected.

Answer: In the present study, we actually employed most available techniques that can measure HBV replication in vitro; levels of HBV proteins by WB and IFA, levels of extracellular HBV particles by conventional PCR and qPCR, levels of HBeAg by ELISA, and levels of intracellular cccDNA by Southern blotting. We cannot find any reason that we should detect HBV RNA levels for some experiments.    

5. Figure 6, the quantification is needed.

Answer: To answer the reviewer’s comment, WB data in Fig. 6B were quantified and shown in Fig. 1C. WB data in Fig. 6A was already quantified in the original manuscript. The legend for Fig. 1C was newly added.

6. If necessary, human primary hepatocytes could be used.

Answer: Human primary hepatocytes may do not have an advantage over human hepatoma cells such as HepG2-NTCP and Hep3B-NTCP in studying the roles of p53 in HBV replication in vitro, despite of its limitations due to availability, cost, genetic variations, HBV replication efficiency, etc.

Round 2

Reviewer 2 Report

Although these authors answered my questions, I still want to point out that the RNA level and primary cells are essential with following reasons:

1. HBV rcDNA is from pgRNA by reverse transcription, if we don't know the change of HBV RNA, we will don't know p53 inhibits HBV replication on which step. So many steps involved in HBV life cycle.

2. As we know, some HBV experiment data are artificial in HCC cell lines.

Author Response

1. HBV rcDNA is from pgRNA by reverse transcription, if we don't know the change of HBV RNA, we will don't know p53 inhibits HBV replication on which step. So many steps involved in HBV life cycle.

As the reviewer commented, p53 can affect HBV replication by inhibiting HBV transcription either directly or indirectly, which was also supported by some experimental evidence,

1) Lines 788-792: Treatment with PFT-a substantially increased HBV replication in the presence p53 (Figure 3), indicating p53 transcriptional activity is involved, at least in part, in the p53-mediated inhibition of HBV replication.

2) Lines 803-807: As p53 is still able to inhibit the replication of HBx-null HBV (Figure 4), other viral proteins may be involved in the p53-mediated anti-HBV defense. It is also possible to assume that p53 inhibits the replication of HBx-null HBV by directly inactivating the HBV core promoter and enhancer, as previously described (51, 52).

However, the p53-mediated inhibition of HBV transcription is not the main subject of the present study. Instead, we focused on the finding that p53 can inhibit HBV replication via a novel mechanism, i.e. downregulation of HBx, a positive regulator of HBV replication, through proteasomal degradation involving E6AP, as follows;

1) Lines 792-796: The potential of p53 to inhibit HBV replication remained active after treatment with PFT-α, which completely inhibited the ability of p53 to activate the expression of its target genes, such as p21 and PUMA (Figure 3), suggesting that p53 can act as a negative regulator of HBV replication via another mechanism that does not involve its transcriptional activity.

2) Lines 799-803: The role of HBx as a positive regulator of HBV replication was more evident in the absence of p53 (Figure 4). Additionally, the ability of p53 to inhibit replication of HBx-null HBV was relatively lower than its ability to inhibit the replication of WT HBV (Figure 4). These results suggest that p53 inhibits HBV replication by restricting the potential of HBx as a positive regulator of HBV replication.

Based on these two findings and others, we conclude that p53 primarily inhibits HBV replication via downregulation of HBx, a positive regulator of HBV replication. Considering that HBx activates the HBV promoter and enhancers to stimulate HBV transcription, HBV transcription is presumed to be decreased if p53 downregulates HBx levels. Therefore, p53 can inhibit HBV transcription via diverse mechanisms, indicating that measurement of HBV RNA levels does not necessarily provide valuable information on the present study.

2. As we know, some HBV experiment data are artificial in HCC cell lines.

Answer: We agree to the opinion that some biological events in HCC cell lines can be different with those actually occurred in the liver of patients. However, we have two important reasons to employ HCC cell liens as model systems for studying the effect of p53 on HBV replication in the present study.

1) Primary hepatocytes and HepaRG, a hepatic cell system that can produce early hepatic progenitor cells as well as completely mature human hepatocytes, were often used for HBV infection studies simply because of the extremely low efficiencies of HBV replication in HCC cell lines. Recently, however, the HBV infection in cultured HCC lines has been extensively optimized and therefore most labs now employ HepG2-NTCP and Hep3B-NTCP cells as a well-established in vitro HBV replication system, especially for studies on the measurement of HBV replication rates.

2) Human primary hepatocytes may more correctly reproduce the nature of human hepatocytes in human liver. However, it is not commonly employed in HBV replication studies, because of limitations in cost, availability, and more importantly low HBV replication efficiency. In addition, it may take at least two months to complete the requested experiments in primary cells, which will delay our publication. Instead, we wish to publish our paper after changing the title to show that the experiments were done in HCC cell lines as shown in line 4.